# Recurrent predictive coding models for associative memory employing covariance learning

**Mufeng Tang[1], Tommaso Salvatori[2], Beren Millidge[1], Yuhang Song[1,2], Thomas Lukasiewicz[2,3], Rafal Bogacz[1]***

**1** MRC Brain Network Dynamics Unit, University of Oxford, Oxford, United Kingdom, **2** Department of Computer Science, University of Oxford, Oxford, United Kingdom, **3** Institute of Logic and Computation, TU Wien, Vienna, Austria

* rafal.bogacz@ndcn.ox.ac.uk

**Data Availability Statement:** The code that can be used to reproduce the experiments in this work is openly available at https://github.com/C16Mftang/covariance-learning-PCNs.

## Abstract

The computational principles adopted by the hippocampus in associative memory (AM) tasks have been one of the most studied topics in computational and theoretical neuroscience. Recent theories suggested that AM and the predictive activities of the hippocampus could be described within a unitary account, and that predictive coding underlies the computations supporting AM in the hippocampus. Following this theory, a computational model based on classical hierarchical predictive networks was proposed and was shown to perform well in various AM tasks. However, this fully hierarchical model did not incorporate recurrent connections, an architectural component of the CA3 region of the hippocampus that is crucial for AM. This makes the structure of the model inconsistent with the known connectivity of CA3 and classical recurrent models such as Hopfield Networks, which learn the covariance of inputs through their recurrent connections to perform AM. Earlier PC models that learn the covariance information of inputs explicitly via recurrent connections seem to be a solution to these issues. Here, we show that although these models can perform AM, they do it in an implausible and numerically unstable way. Instead, we propose alternatives to these earlier covariance-learning predictive coding networks, which learn the covariance information implicitly and plausibly, and can use dendritic structures to encode prediction errors. We show analytically that our proposed models are perfectly equivalent to the earlier predictive coding model learning covariance explicitly, and encounter no numerical issues when performing AM tasks in practice. We further show that our models can be combined with hierarchical predictive coding networks to model the hippocampo-neocortical interactions. Our models provide a biologically plausible approach to modelling the hippocampal network, pointing to a potential computational mechanism during hippocampal memory formation and recall, which employs both predictive coding and covariance learning based on the recurrent network structure of the hippocampus.

**Funding:** This work has been supported by Medical Research Council UK grant MC_UU_00003/1 to RB, and an E.P. Abraham Scholarship in the Chemical, Biological/Life and Medical Sciences to MT. The funders had no role in study design, data collection and analysis, decision to publish, or preparation of the manuscript.

## Author summary

The hippocampus and adjacent cortical areas have long been considered essential for the formation of associative memories. It has been recently suggested that the hippocampus stores and retrieves memory by generating predictions of ongoing sensory inputs. Computational models have thus been proposed to account for this predictive nature of the hippocampal network during associative memory using *predictive coding*, a general theory of information processing in the cortex. However, these hierarchical predictive coding models of the hippocampus did not take into account important hippocampal architectural components that may store statistical regularities supporting associative memory, which also hinders a unified view with classical models learning these regularities. To address these issues, here we present a family of predictive coding models that also learn the statistical information needed for associative memory. Our models can stably perform associative memory tasks in a biologically plausible manner, even with large structured data such as natural scenes. Our work provides a possible mechanism of how the recurrent hippocampal network may employ various computational principles concurrently to perform associative memory.

## Introduction

Memory systems in the brain often store information about the relationships or associations between objects or concepts. This particular type of memory, referred to as *Associative Memory* (AM), is ubiquitous in our everyday lives. For example, we memorize the smell of a particular brand of perfume, the taste of a kind of coffee, or the voice of different singers we like. After a memory is formed, AM will support its retrieval when a related cue is presented to our senses using the learned association between the provided cues and missing components.

It has long been argued that the hippocampus and adjacent cortical areas, located in the medial temporal lobe of the brain, are crucial for AM [1–3]. Historically, various theoretical and computational works have been developed in an effort to model the hippocampus in AM tasks [4–6]. In this work, we are interested in a particular approach to modelling, which assumes that the hippocampus employs its *predictive activities* to perform AM. Predictive processing is thought to be a key computational principle underlying various hippocampal activities: Experimentally, abundant evidence has suggested that the hippocampus is capable of predicting ongoing sensory inputs [7], whereas high-level theories and computational models have also been proposed to explain how predictive coding (PC) may support various properties of the hippocampus, including the formation of cognitive maps [8] and memory [9]. In particular, Barron et al. [9] proposed that the hippocampus sits at the top of a hierarchical generative model that generates predictions of neocortical activities based on past experiences, thus enabling retrieval of activities from memory at lower neocortical levels. The ability of predictive coding networks (PCNs) to complete previously learned patterns have been already demonstrated in pioneering work by Rao [10], although the model in this work was not formally introduced within the PC framework. More recently, the ability of the hierarchical PCNs introduced by Rao and Ballard [11] in performing AM has been computationally analysed [12, 13]. Specifically, Salvatori et al. [12] showed that hierarchical PCNs can store training data points as memories, and retrieve these memories given partial or noisy cues. Under this PC framework, which is characterized by prediction error neurons, the memorization of a sensory input is driven by the Hebbian learning dynamics [14] that minimize the error between the input and a prediction generated by the network, and the retrieval of this input is performed by the

inferential neural dynamics, also to minimize the error between the internal predictions and the corrupted sensory input. It is worth mentioning that AM in this PC-based model is defined as the memorization and recall of *static* inputs, where the associations are learned between individual components of static patterns, and the temporal dimension is ignored. In this scenario, memory retrieval is equivalent to pattern completion. On the other hand, AM can also be defined as memorizing the association along the temporal dimension, where the associations between two (or multiple) inputs along time are memorized [10]. In this work we focus on the first i.e., static type of AM.

The aforementioned hierarchical PC-based model provides a possible network mechanism that the hippocampo-neocortical entity employs to support AM, providing a possible implementation of the theory by Barron et al. [9]. However, this model only included feedforward and feedback connections between layers of neurons (representing different cortical areas), and thus failed to provide an account of how PC can also be employed within the known connectivity of the hippocampal network, where the *recurrent* connections in subfield CA3 are thought to play a key role in AM [5]. Furthermore, the absence of recurrent connections in the hierarchical PC models for AM dissociates them from earlier recurrent models of AM such as Hopfield Network [4], which assume that the recurrent connections in the hippocampal network learns a covariance matrix representing the association between individual neurons activated by a memory item, thus hindering a unified understanding of the computational principles adopted by the hippocampus to support AM. This brings us to ask: **how can the recurrent hippocampal network employ PC to perform AM, in a biologically plausible and computationally stable manner, and can recurrent connections in such network encode the covariance of neural activity?** Here, by computationally stable, we mean the ability of the model to steadily converge to a fixed point both during learning (memory) and inference (retrieval), and the criteria for biological plausibility include [15]:

1. *Local computation*: A neuron's computation is only dependent on its input neurons and weights connecting itself to its input neurons.

2. *Local plasticity*: The plasticity rule of synapses in a model only depends on quantities encoded by pre- and post-synaptic neurons.

3. *Architectural similarity*: Components of a model resembles architectures of real neurons, such as the recurrent connections and the apical dendrites of the pyramidal neurons in the hippocampus.

The hierarchical PC models for AM are stable and satisfy the first two criteria of plausibility [12]. However, as we pointed out above, they fail to meet the third criterion, architectural similarity, due to the missing recurrent connections. In this work, we propose a family of PCNs with recurrent connections between neurons, which we call covariance-learning PCNs (covPCNs), as candidate models satisfying these criteria. In particular, we first identify that an earlier type of PCN has already incorporated recurrent synaptic connections encoding the covariance information of inputs [16–18], and can thus be considered as a PC model meeting the criterion of architectural similarity. We refer to it as the *explicit covPCN*, as it encodes the covariance matrix explicitly into its recurrent synapses. The explicit covPCN was originally proposed as a model for learning representations of sensory inputs, and we show in this work that its covariance-learning nature can be utilized to perform simple AM tasks. However, we note that the learning rule for the recurrent connections in this model is non-Hebbian, and poses significant computational issues as well, which makes it fail to satisfy the local plasticity and stability conditions. To address these issues, we propose in this work a novel recurrent PCN, which also encodes the covariance matrix via its recurrent connections, but in an

implicit manner, thus we refer to it as *implicit covPCN*. We show that the new implicit model also performs AM via covariance learning, and it is equivalent to the explicit covPCN both theoretically and empirically in simple AM tasks, while only employing local Hebbian plasticity. We also show that the implicit model can be further modified to achieve biological resemblance to the hippocampal pyramidal cells by incorporating a dendritic structure, while retaining the theoretical and empirical equivalence to the explicit covPCN at convergence. We name it the *dendritic covPCN* in this work. Importantly, we show that both the implicit and dendritic models can perform more complex AM tasks in which the explicit covPCN would fail due to its unstable dynamics. Finally, we propose a *hybrid PCN* that combines the implicit covPCN with a hierarchical PCN [11, 12] to model the whole hippocampo-neocortical region, and show that it performs challenging AM tasks efficiently.

In this work, we describe a series of more stable and more plausible implementations of recurrent PC to model AM in the hippocampus, to shed light on how the hippocampal *structure* (apical dendrite and recurrent network) may support its *computations* (PC and covariance learning) underlying its *functions* (AM). The key contribution that we bring to the table is a reparameterization of the (weighted) prediction errors in the explicit covPCN [16], which leads to simplified forms of a free energy. Crucially, the simplified forms of free energy that we consider all share the same minima or fixed points with that of the explicit covPCN, thereby leading to the same inference and learning. Practically, this allows us to drop certain terms from the gradients, leading to more robust convergence to free energy minima and, crucially, affording more biologically plausible implementation. To unpack the basic ideas of how PC can be implemented in CA3-like recurrent network to support AM, we will focus on a simple kind of PC, in which we ignore temporal predictions (i.e., predicting into the future). Rather, in our models, the prediction of one neuron's activities is from all other neurons in the recurrent network i.e., "spatial" prediction. This particular focus enables us to derive interpretable analytical results that enhance our understanding of what is encoded in the synaptic weights in our models i.e., the covariance matrix and the relationship between models. Furthermore, we will restrict our initial analyses to linear systems, under standard Gaussian parametric assumptions. In this setting, a memory is simply the ability of the PCN to recognize the most likely cause of a particular pattern of inputs (e.g., an image), which is similar to the AM tasks discussed in [12]. We are not addressing episodic memory—of the sort sometimes associated with hippocampal function—rather, we are focusing on how statistical regularities in a series of inputs are learned and then used to predict the missing or noisy part of a input. This contrasts our linear recurrent PCNs with recurrent AM models such as the Hopfield Network [4], where the memories are stored as point attractors of the network dynamics. At the end of the Results section, we provide results of an empirical analysis of the attractor behavior of our model, showing that adding nonlinearities to our model will enable it to store memories as point attractors.

## Models

In this section, we introduce the single-layer covariance-learning PCNs, i.e., the explicit, implicit and dendritic covPCNs, following increasing levels of biological plausibility. To perform AM tasks, covPCNs first memorize a set of patterns by learning their model parameters to minimize an objective function, or more specifically, (variational) free energy [16] via gradient descent. After memorization, the covPCNs are given a set of cues, such as corrupted memories, and they will perform retrieval by performing inference or relaxation of the neurons to achieve minimization of the energy function again. We focus on how computations are carried out in these models and their corresponding neural implementations, which aim to model the

recurrent networks in the sub-field CA3 of hippocampus. We then describe a full model for the hippocampo-neocortical region as a whole, which uses the implicit/dendritic covPCNs to model the hippocampal recurrent network and the hierarchical PCN [12] to model the neocortical hierarchy.

### Explicit covariance-learning PCNs

The explicit covPCNs [16, 17] were proposed to extend the PC model for the visual system [11] to a general model of how the brain performs representation learning. Following the probabilistic framework of PC, it introduced the covariance matrix by encoding it explicitly into the network's recurrent connections. We denote the activity of neurons in a single-layer explicit covPCN by a vector $\mathbf{x}$, and throughout the paper we denote vectors with a bold font. The model assumes that the set of patterns to memorize $\{\mathbf{x}(i)\}_{i=1}^{N}$ of dimension $d$ is a set of samples from a Gaussian distribution with true mean $\boldsymbol{\mu}_{\text{true}}$ and covariance matrix $\Sigma_{\text{true}}$, where $i$ indicates the $i$th sample within the dataset. The dynamics of the network model, parameterized by mean $\boldsymbol{\mu}$ and covariance $\Sigma$, thus aim to maximize the following log likelihood of observed neural activity $\mathbf{x}$ given this Gaussian distribution (or negative free energy):

$$\mathcal{F} = -\frac{1}{2}\log|\Sigma| - \frac{1}{2}(\mathbf{x} - \boldsymbol{\mu})^{T}\Sigma^{-1}(\mathbf{x} - \boldsymbol{\mu}) \tag{1}$$

An explicit covPCN learns its parameters by iteratively setting neural activity to training patterns $\mathbf{x} = \mathbf{x}(i)$ and then modifying parameters by directly computing the derivative of $\mathcal{F}$ with respect to $\boldsymbol{\mu}$ and $\Sigma$ [17]:

$$\boldsymbol{\mu} \leftarrow \boldsymbol{\mu} + \Delta\boldsymbol{\mu} = \boldsymbol{\mu} + \alpha\frac{\partial\mathcal{F}}{\partial\boldsymbol{\mu}}\bigg|_{\mathbf{x}=\mathbf{x}(i)}, \quad \Sigma \leftarrow \Sigma + \Delta\Sigma = \Sigma + \alpha\frac{\partial\mathcal{F}}{\partial\Sigma}\bigg|_{\mathbf{x}=\mathbf{x}(i)} \tag{2}$$

where $\alpha$ denotes the learning rate for parameters. Notice that this learning rule is fully online as it updates the parameters every time a single training pattern is received. This is closer to learning by biological systems. Here, to derive subsequent analytical properties of the covPCNs, we define the following full-batch learning rules, which can approximate the fully online learning if $\alpha$ is small enough:

$$\Delta\boldsymbol{\mu} = \alpha\sum_{i=1}^{N}\frac{\partial\mathcal{F}}{\partial\boldsymbol{\mu}}\bigg|_{\mathbf{x}=\mathbf{x}(i)} = \alpha\sum_{i=1}^{N}\Sigma^{-1}(\mathbf{x}(i) - \boldsymbol{\mu}) \tag{3}$$

$$\Delta\Sigma = \alpha\sum_{i=1}^{N}\frac{\partial\mathcal{F}}{\partial\Sigma}\bigg|_{\mathbf{x}=\mathbf{x}(i)} = \alpha\left(-N\Sigma^{-1} + \sum_{i=1}^{N}\Sigma^{-1}(\mathbf{x}(i) - \boldsymbol{\mu})(\mathbf{x}(i) - \boldsymbol{\mu})^{T}\Sigma^{-1}\right) \tag{4}$$

The above full-batch learning rules have a property that both parameters will converge to the maximum likelihood estimate (MLE) of $\boldsymbol{\mu}_{\text{true}}$ and $\Sigma_{\text{true}}$ based on the training data points, i.e., $\boldsymbol{\mu} \rightarrow \frac{1}{N}\sum_{i=1}^{N}\mathbf{x}(i)$ and $\Sigma \rightarrow \frac{1}{N}\sum_{i=1}^{N}(\mathbf{x}(i) - \boldsymbol{\mu})(\mathbf{x}(i) - \boldsymbol{\mu})^{T}$. We denote these MLE estimates of mean and covariance by $\bar{\mathbf{x}}$ and $S$, respectively. Therefore, the parameter matrix $\Sigma$ will *explicitly* encode the sample covariance of the data $S$, thus the name explicit covPCNs. This can be shown by noting that at convergence $\boldsymbol{\mu}$ and $\Sigma$ do not change, so setting $\Delta\boldsymbol{\mu} = 0$ and $\Delta\Sigma = 0$ and solving Eqs 3 and 4 for $\boldsymbol{\mu}$ and $\Sigma$, respectively, gives the above MLE estimates. It is worth noting that although we refer to these estimates MLE, in more general formulations they would correspond to maximum a posteriori (MAP) estimates [16, 18]. However, because we have not

placed any prior constraints on the generative model implied by Eq 1, the MLE and MAP become equivalent.

After learning, we fix the parameters $\boldsymbol{\mu}$ and $\boldsymbol{\Sigma}$ and provide a single cue $\tilde{\mathbf{x}}$ to the network, i.e., we initialize the neural activity to $\mathbf{x} = \tilde{\mathbf{x}}$, and the explicit covPCN performs inference on the cue by updating it according to the derivative of the log likelihood:

$$\Delta \mathbf{x} = \beta \frac{\partial \mathcal{F}}{\partial \mathbf{x}} = \beta\left(-\boldsymbol{\Sigma}^{-1}(\mathbf{x} - \boldsymbol{\mu})\right) \tag{5}$$

where $\beta$ defines step size along the gradient direction, which we refer to as "integration step". For example, if $\tilde{\mathbf{x}}$ is a corrupted or noisy data point, the equation above will drive it towards the mean in proportion to the precision or inverse variance.

To see how the above equations could be implemented in a neural circuit, it is useful to note that they greatly simplify if we define a vector of prediction errors:

$$\boldsymbol{\varepsilon} = \boldsymbol{\Sigma}^{-1}(\mathbf{x} - \boldsymbol{\mu}) \tag{6}$$

Then, the dynamics of neurons (Eq 5) becomes:

$$\Delta \mathbf{x} = -\beta \boldsymbol{\varepsilon} \tag{7}$$

Moreover, the learning rules for this model (Eqs 3 and 4) become:

$$\Delta \boldsymbol{\mu} = \alpha \sum_{i=1}^{N} \boldsymbol{\varepsilon}(i) \tag{8}$$

$$\Delta \boldsymbol{\Sigma} = \alpha \left( -N \boldsymbol{\Sigma}^{-1} + \sum_{i=1}^{N} \boldsymbol{\varepsilon}(i)\boldsymbol{\varepsilon}(i)^T \right) \tag{9}$$

The above neural dynamics (Eqs 6–9) can be implemented within the network shown in Fig 1A. In this network, $\boldsymbol{\varepsilon}$ and $\mathbf{x}$ are encoded in activities of neurons, and parameters $\boldsymbol{\mu}$ and $\boldsymbol{\Sigma}$

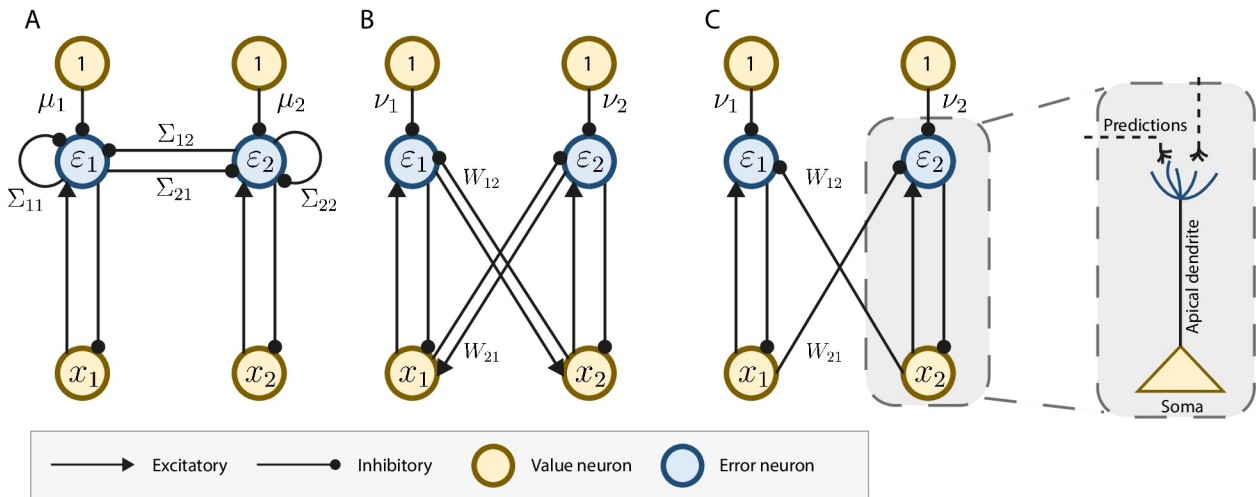

**Fig 1. Neural implementations of the single-layer covPCNs using error neurons.** $x_a$ and $\varepsilon_a$ denotes the $a$-th value/error neuron in the networks. For simplicity we show the case where the input patterns have only 2 dimensions, corresponding to 2 value and error neurons. A: explicit covPCN originally proposed in [16]. B: Implicit covPCN. C: Dendritic covPCN with direct structural mapping to a pyramidal cell. Unlabeled connections have strengths 1.

in synaptic weights. The neurons **x**, called the value neurons, receive inhibition from prediction error neurons (cf. Eq 7), while the neurons $\varepsilon$ receive excitatory inputs from the value neurons, and top-down inhibitory inputs encoding the prior expectation, as well as lateral inhibitory inputs encoding the weight $\Sigma$, to compute the weighted prediction error of Eq 6 (for details of how this computation arises see [18]).

Notice that the learning for the top-down connections encoding the predictions $\mu$ (Eq 8) follows the Hebbian rule, as it is simply the product of the pre-synaptic activity (1) and post-synaptic error activity ($\varepsilon$). However, the learning rule for the lateral connections $\Sigma$ (Eq 9) is biologically implausible, since to compute $\Sigma_{ab}^{-1}$ that is needed for the update $\Delta\Sigma_{ab}$ of the synapse connecting neuron $a$ and $b$, non-local information from all other entries in $\Sigma$ is needed, due to the nature of the inverse operation. That is, to compute the synaptic update between neuron $a$ and $b$, synaptic weights from all over the neural circuit are needed. Moreover, as we will show in the Results section, this numerically unstable inverse term poses significant computational problems that make this model inapplicable to complex patterns, such as the image datasets MNIST [19] and CIFAR10 [20], which are not generated by sampling from a Gaussian distribution.

## Implicit covariance-learning PCNs

The biological implausibility of the explicit covPCNs raises the question whether there exists an alternative and more realistic PC model that performs AM via covariance learning. Notice that another way of encoding interactions between neurons, while preserving the predictive nature of the network, is to let the neurons predict each other. With this intuition, we propose the implicit covPCN, which is also a recurrently connected, single-layer network with weight matrix $W$, with error neurons encoding the prediction errors. The implicit model aims to maximize the following negative energy function:

$$\mathcal{F} = -\frac{1}{2}\left\|\mathbf{x} - W\mathbf{x} - v\right\|_2^2 \tag{10}$$

where **x** and **v** are both $d$-dimensional vectors and $W$ is a $d \times d$ 0-diagonal matrix, where $W_{ab}$ will be encoded in the synapse connecting the $a$-th and $b$-th neurons in the recurrent network. In the implicit model, we define the errors as $\varepsilon = \mathbf{x} - W\mathbf{x} - v$. The 0-diagonal property of $W$ implies that the predictions $W\mathbf{x}$ come from all value neurons in the vector of neurons **x** except each neuron itself. Like the explicit model, the implicit covPCN first updates its parameters $W$ and $v$ by computing the derivative of $\mathcal{F}$ given the dataset $\{\mathbf{x}(i)\}_{i=1}^N$ (again, we present the full-batch learning rules here for subsequent analytical derivations):

$$\Delta v = \alpha \sum_{i=1}^N \left.\frac{\partial \mathcal{F}}{\partial v}\right|_{\mathbf{x}=\mathbf{x}(i)} = \alpha \sum_{i=1}^N \varepsilon(i) \tag{11}$$

$$\Delta W = \alpha \sum_{i=1}^N \left.\frac{\partial \mathcal{F}}{\partial W}\right|_{\mathbf{x}=\mathbf{x}(i)} = \alpha \left(\sum_{i=1}^N \varepsilon(i)\mathbf{x}(i)^T\right)_{diag=0} \tag{12}$$

where the $()_{diag=0}$ notation means "enforcing the diagonal elements to be 0" as we want to keep the 0 diagonal elements of $W$ unchanged. Notice that by setting $\Delta v$ and $\Delta W$ to 0, we obtain the

following relationships at the convergence of learning:

$$\boldsymbol{v} = (I - W)\bar{\mathbf{x}} \tag{13}$$

$$[(I - W)S]_{diag=0} = 0 \tag{14}$$

Recall that $\bar{\mathbf{x}}$ and $S$ denote the MLEs of the mean $\boldsymbol{\mu}_{true}$ and covariance matrix $\Sigma_{true}$, and that the parameters of the explicit covPCN, $\boldsymbol{\mu}$ and $\Sigma$, converge to $\bar{\mathbf{x}}$ and $S$. Therefore, the implicit covPCN also learns the mean and the elements of the covariance matrix, without encoding them explicitly in individual connections.

Like the explicit model, after learning, the implicit model performs inference after initializing the activity to a cue input $\tilde{\mathbf{x}}$ following the derivative of the objective function with respect to $\mathbf{x}$:

$$\Delta\mathbf{x} = \beta\frac{\partial\mathcal{F}}{\partial\mathbf{x}} = \beta(-\boldsymbol{\varepsilon} + W^T\boldsymbol{\varepsilon}) \tag{15}$$

The above dynamics can be implemented in the network model in Fig 1B, by replacing the lateral connections $\Sigma$ with $W$ that projects the predictions from all other value neurons into each error neuron. Notice that the learning rule for the bias term $\boldsymbol{v}$ is Hebbian, and more importantly, the learning rule for the connections $W$ is also Hebbian, as it is simply a product $\boldsymbol{\varepsilon}\,\mathbf{x}^T$ of the pre- and post-synaptic activities. We show in the Results section, that this biologically more plausible implicit model will converge to exactly the same retrieval of a memory as the explicit model, making it a perfect alternative to the implausible explicit model.

## Dendritic covariance-learning PCNs

Inspired by the dendritic model in [21, 22], we push the biological plausibility of the implicit model further by imposing a "stop gradient" (sg) operation on the objective function Eq 10:

$$\mathcal{F} = -\frac{1}{2}\left\| \mathbf{x} - W\mathrm{sg}(\mathbf{x}) - \boldsymbol{v} \right\|_2^2 \tag{16}$$

where

$$\mathrm{sg}(x) = x; \quad \frac{\partial\mathrm{sg}(x)}{\partial x} = 0 \tag{17}$$

Notice that the parameter learning rules in this model is exactly the same as Eqs 13 and 14. Only the dynamics of the value nodes during inference becomes:

$$\Delta\mathbf{x} = -\beta\boldsymbol{\varepsilon} \tag{18}$$

The above dynamics differs from the implicit model (Eq 15) that it no longer includes $W^T\boldsymbol{\varepsilon}$ term that corresponded to the backward excitatory connections from the error neurons to the value neurons in Fig 1B. This results in the neural network implementation shown in Fig 1C. This implementation is simpler, as the error neurons in this model only receive predictions from external sources, so they can thus be absorbed into the dendrites of the value neurons $x$, as shown in the soma-dendrite demonstration in Fig 1C. This architecture could be viewed as more biologically plausible because it does not include the special one-to-one connections of strength 1 between value and corresponding error neurons present in Fig 1A and 1B, for which there is no evidence in cortical circuits. Such an error-encoding dendritic model relates to the model proposed in [21, 22]. As we show in the Results section, since this stop gradient operation does not affect the learning of parameters in the implicit covPCN, and the fixed

points of dynamics for the implicit and dendritic models are the same, the equivalence to the explicit model in converged memory retrieval is retained.

## Hybrid PCNs

We now use the obtained results to propose an architecture that mimics the behavior of the hippocampus as a memory index and generative model. The recurrent, single-layer network models introduced above provide a model for the recurrent dynamics in the hippocampus. However, raw sensory inputs are first processed via a hierarchy of sensory and neocortical layers, allowing the hippocampus to memorize representations of raw signals, instead of the signal itself. Moreover, according to a recent theory, the hippocampus functions as a generative model that accumulates prediction errors from sensory and neocortical neurons lower in the hierarchy, and sends descending predictions to the neocortex, to correct the prediction errors in the neocortical neurons. The generative model of the hippocampus is updated until the hippocampal predictions correct the prediction errors by suppressing neocortical activity [9]. Hierarchical PCNs have already been proposed as a generative model that learn representations of the sensory inputs [17]. Particularly, it has been shown that a purely hierarchical PCN, without any recurrent structure, is able to perform associative memory tasks on highly complex datasets [12]. Here, we combine the hierarchical PCN with our proposed recurrent architecture, obtaining an hierarchical model with recurrent dynamics at the topmost layer. What results is an *hybrid* network that models the whole pathway from sensory neurons to hippocampal neurons.

Consider an hierarchical PCN with $L$ layers, with a recurrent implicit or dendritic network connected to the last layer. The first layer corresponds to the sensory layer, where sensory signals are presented to be processed, and the last layer corresponds to the hippocampal layer. Then, the intermediate layers $1 < l < L$ represent the hierarchical structure present in the neocortex that connects the sensory neurons to the hippocampus. We denote the synaptic weight matrix connecting the neurons in the $l$th layer and the neurons in the $(l + 1)$th layer as $\Theta^{(l)}$. Within the hippocampal layer $L$, we use $W$ to denote the recurrent synaptic weight matrix. The vector of value nodes in each layer, denoted as $\mathbf{x}^{(l)}$, is coupled with the error node vector $\boldsymbol{\varepsilon}^{(l)}$, computed as the following prediction error:

$$\boldsymbol{\varepsilon}^{(l)} = \mathbf{x}^{(l)} - \boldsymbol{\rho}^{(l)} \tag{19}$$

where $\boldsymbol{\rho}^{(l)}$ denotes the descending prediction signals into the neurons, computed as:

$$\boldsymbol{\rho}^{(l)} = \begin{cases} W\mathbf{x}^{(l)}, & \text{if } l = L \text{ and implicit} \\ W\mathrm{sg}(\mathbf{x}^{(l)}), & \text{if } l = L \text{ and dendritic} \\ \Theta^{(l)}f(\mathbf{x}^{(l+1)}), & \text{otherwise} \end{cases} \tag{20}$$

where $f(\cdot)$ is a nonlinear function. For simplicity of illustration, we ignore the top-down connections $\boldsymbol{v}$ in the recurrent covPCNs. The aim of the learning dynamics of this hybrid network is to maximize the negative sum of squared errors:

$$\mathcal{F} = -\frac{1}{2}\sum_{l=1}^{L} \left\| \boldsymbol{\varepsilon}^{(l)} \right\|_2^2 \tag{21}$$

Like purely hierarchical PCNs, the learning of the hybrid PCN consists of two stages: *inference* and *weight update*. During inference, the value neurons relax to maximize $\mathcal{F}$ following

the gradient ascent rule:

$$\Delta\mathbf{x}^{(l)} = \begin{cases} \beta\big(-\boldsymbol{\varepsilon}^{(l)} + f'(\mathbf{x}^{(l)}) \odot (\Theta^{(l-1)})^T \boldsymbol{\varepsilon}^{(l-1)} + \gamma W^T \boldsymbol{\varepsilon}^{(l)}\big), & \text{if } l = L \\[2mm] \beta\big(-\boldsymbol{\varepsilon}^{(l)} + f'(\mathbf{x}^{(l)}) \odot (\Theta^{(l-1)})^T \boldsymbol{\varepsilon}^{(l-1)}\big), & \text{if } 1 < l < L \\[2mm] \mathbf{0}, & \text{if } l = 1 \end{cases} \tag{22}$$

where $\odot$ denotes the element-wise product between two vectors, and $\gamma = 1$ if the topmost layer is an implicit covPCN, or 0 if it is a dendritic covPCN. The update at the sensory layer ($l = 1$) is 0 because during learning, the sensory neurons are fixed at the raw sensory inputs. The inference dynamics are carried out for a certain number of iterations until the value nodes reach an equilibrium. The weight update is then performed, also to maximize $\mathcal{F}$ using gradient ascent:

$$\Delta\Theta^{(l)} = \alpha\boldsymbol{\varepsilon}^{(l)} f(\mathbf{x}^{(l+1)})^T \quad \text{and} \quad \Delta W = \alpha\boldsymbol{\varepsilon}^{(L)}(\mathbf{x}^{(L)})^T \tag{23}$$

In each iteration of learning, the inference will be performed for multiple iterations and then the weight update will take place for one step. The whole learning process will be iterated multiple times until $\mathcal{F}$ reaches the minimum.

The proposed architecture also models the reconstructions of past memories: as a memory index, the hippocampus sends top-down information to the neocortical neurons to reinstate activity patterns that replicate previous sensory experience [9]. In the reconstruction phase of our generative model, the hippocampal layer provides descending inputs to the sensory neurons to generate stored data points. Assume we have a hybrid PCN already trained on a dataset of memories. To retrieve a stored datapoint, the synaptic weights $\Theta$ and $W$ are fixed, and the retrieval process is triggered by providing a partly corrupted version of a memorized pattern to a subset sensory neurons. During retrieval all other value nodes will relax following the same rule specified in Eq 22. The only difference is that, the value nodes in the sensory layer will also experience relaxation:

$$\Delta\mathbf{x}_c^{(1)} = -\beta\boldsymbol{\varepsilon}_c^{(1)} \tag{24}$$

where the subscript $c$ denote the corrupted dimensions of the input pattern, as we keep the intact part of the patterns unchanged during retrieval. Notice that all the dynamics above requires only local computations. Fig 2 shows the general structure of the hybrid PCN, as well as detailed implementations of neural computations.

## Results

In this section we present key findings with the aforementioned models. We first show that the explicit covPCN performs AM in practice with randomly generated patterns, a property never previously demonstrated for this model. We then derive an expression for retrieved pattern in this model, providing a deeper understanding of its performance in AM tasks. We then show analytically that the retrieved pattern in the implicit/dendritic covPCNs is exactly the same as that of the explicit model when the inferential dynamics have converged, suggesting that these models serve as perfect substitutes for the explicit model, while being biologically more plausible. Moreover, we show that these more plausible models are also more stable with structured image data: they perform AM well on images of handwritten digits and natural objects, whereas the explicit model fails on these datasets due to the inverse term discussed above. We next show that the hybrid model taking into account the hippocampo-neocortical interactions successfully performs AM in a parameter-efficient manner. Finally, we investigate a nonlinear

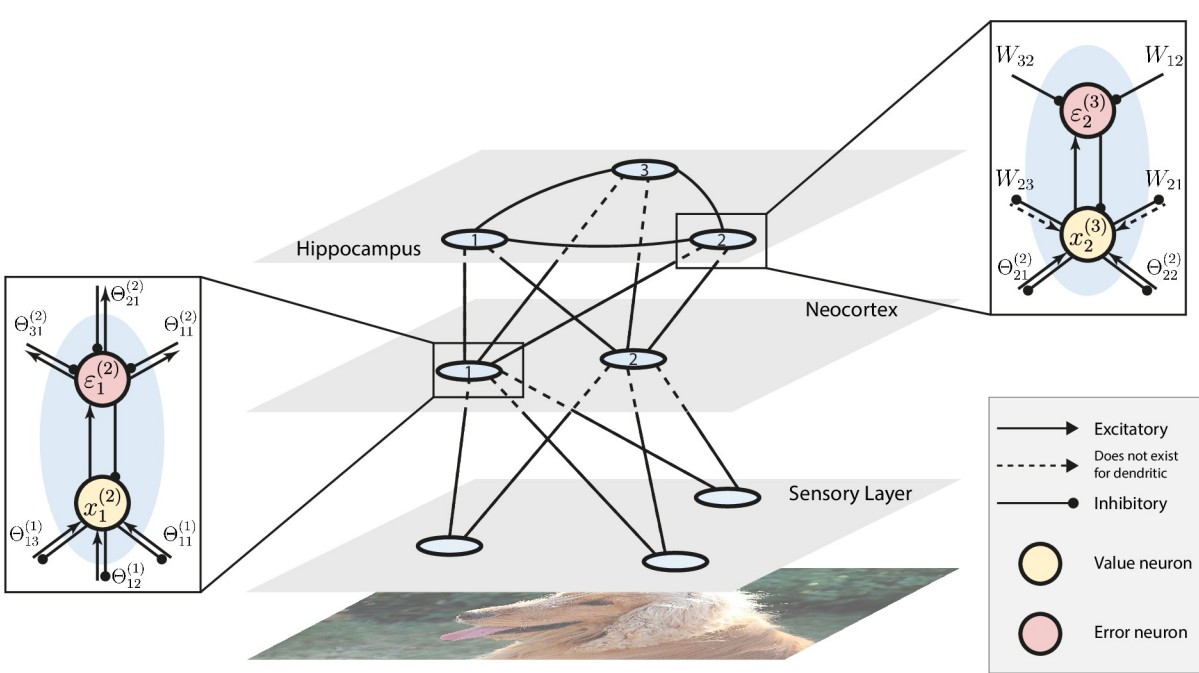

**Fig 2. Multilayer hybrid PCN.** We use the single-layer implicit/dendritic covPCN to model the hippocampus, and a hierarchical PCN from [12] to model the sensory cortex and neocortex. Neurons and synapses in the hierarchical layers follow the dynamic rules in [12]. For clarity of demonstration, only one layer of our neocortex model is shown. Expanded boxes show the detailed computations within individual neurons and related synapses specified in Eqs 20 and 22, where $x_a^{(l)}$ denotes the $a$-th neuron in the $l$th layer, and $\Theta_{ab}$ and $W_{ab}$ denote the individual weights from the $a$th to the $b$th neurons. Dog image in this figure is obtained from Wikimedia Commons under a CC BY 4.0 license.

version of our implicit model and show that they learn individual point attractors of memory. The implementation details of these experiments are provided in S1 and S2 Tables.

### The explicit covPCN performs associative memory

The explicit covPCN was originally proposed as a generative model of how the cortical responses are evoked given observations [16, 17]. To our knowledge, it has never been employed to perform AM tasks. To verify our hypothesis that after training, the explicit covPCN can perform AM using its inferential dynamics and learned covariance, we designed a simple task where multiple $5 \times 5$ random Gaussian patterns are memorized. We then covered the bottom two rows of the $5 \times 5$ patterns with 0s and run inference on these corrupted entries only. Examples of the original patterns, along with the corrupted and retrieved patterns, can be found in Fig 3A. In this task, the models can be considered as memorizing the association between the top 3 rows and bottom 2 rows of the random patterns. As can be seen in the third column of panel A, the single-layered explicit covPCN can retrieve the covered part of the original pattern well: it performs AM. We then measured the MSEs between the retrieved and original $5 \times 5$ random patterns by these models, and plotted them as a function of corrupted/masked pixels in Fig 3B (Fig 3A shows examples when there are 10 masked pixels). As can be seen, the explicit covPCN has slightly higher retrieval MSEs on average, although this performance gap is not visually observable from the retrieved patterns (e.g., Fig 3A). As we will show in the following sections, this performance gap will be exacerbated when these models are trained to memorize structured and more complex datasets.

We now show that a theoretical result can be derived to analytically describe the pattern retrieved by an explicit covPCN. We will use this result later to establish equivalence between

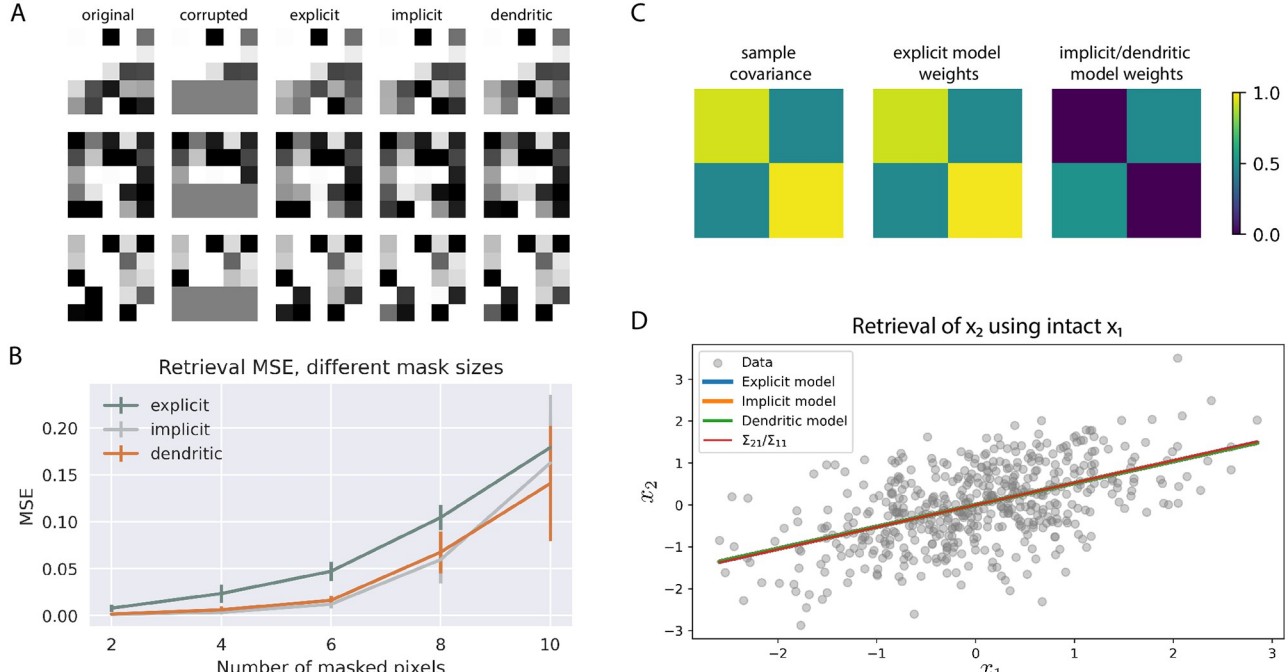

**Fig 3. Performance of covPCNs in AM of random patterns, and the equivalence between them.** A: A subset of $5 \times 5$ random patterns memorized by all 3 models. After training, we corrupted the bottom 2 rows (10 pixels) and let the networks run inference on the corrupted parts for retrieval. B: Retrieval MSEs of the models when corrupted with different mask sizes. Experiments in A and B are performed with networks with $d = 25$ neurons. C: Sample covariance of a random 2-dimensional dataset and the learned weight matrices of an explicit model and an implicit/dendritic model on this dataset. D: The random 2-dimensional dataset to memorize, and the linear retrieval obtained by masking the second dimension $x_2$ by all 3 models, as well as the theoretical retrieval line. All the lines overlap as they are equivalent in theory. Experiments in C and D are performed with networks with $d = 2$ neurons.

explicit and implicit covPCNs. Consider a corrupted pattern $\mathbf{x} \in \mathbb{R}^d$, which can be divided into the corrupted part $\mathbf{x}_k \in \mathbb{R}^k$ and "intact" part $\mathbf{x}_m \in \mathbb{R}^m$, where $k + m = d$:

$$\mathbf{x} = \begin{bmatrix} \mathbf{x}_k \\ \mathbf{x}_m \end{bmatrix} \tag{25}$$

At the time of retrieval, the training of the covPCN has already finished, during which the network has memorized the association between $\mathbf{x}_k$ and $\mathbf{x}_m$. We assume that the training has converged, so that the connections $\boldsymbol{\mu}$ and $\Sigma$ explicitly encode the MLE, $\bar{\mathbf{x}}$ and $S$, of the mean and covariance respectively. In this case, the inference dynamics of the explicit model follows $\Delta \mathbf{x} = \beta S^{-1}(\mathbf{x} - \bar{\mathbf{x}})$ (Eq 5). To correspond to the division of the corrupted and intact parts of $\mathbf{x}$, we also divide $S$ and $\bar{\mathbf{x}}$ into blocks and set $\Delta \mathbf{x} = 0$ to study the convergence of the inference stage:

$$S = \begin{bmatrix} S_{kk} & S_{km} \\ S_{mk} & S_{mm} \end{bmatrix}, \quad \bar{\mathbf{x}} = \begin{bmatrix} \bar{\mathbf{x}}_k \\ \bar{\mathbf{x}}_m \end{bmatrix} \tag{26}$$

where $S_{pq}$, $p, q \in \{k, m\}$ denotes the $p \times q$ submatrices of the covariance matrix $S$. Using Eq 26 above, we can have the following theorem:

**Theorem 1** *After training, an explicit covPCN retrieves the corrupted* $\mathbf{x}_k$ *using the intact* $\mathbf{x}_m$ *following the dynamics* $\Delta\mathbf{x} = \beta S^{-1}(\mathbf{x} - \bar{\mathbf{x}})$, *and the retrieval dynamics on* $\mathbf{x}_k$ *converge to:*

$$\hat{\mathbf{x}}_k = S_{km}S_{mm}^{-1}(\mathbf{x}_m - \bar{\mathbf{x}}_m) + \bar{\mathbf{x}}_k \tag{27}$$

where $\hat{\mathbf{x}}$ denotes the retrieval of the corrupted part. This theorem describes the activities in the network after retrieval given by the explicit covPCN, using the learned parameters and the intact $\mathbf{x}_m$. Details of the derivation can be found in S1 Appendix.

## The equivalence between explicit and implicit covPCNs

Having established the theoretical foundation of the retrieval dynamics in the explicit covPCN, we next consider the theoretical aspects of the implicit and dendritic models: what is the retrieval of memorized patterns given by these models, based on learned associations? Again consider the case of a vector $\mathbf{x}$ consisting of the top $k$ corrupted entries $\mathbf{x}_k$ and bottom $m$ intact entries $\mathbf{x}_m$, and assume that the learning has converged when the corrupted $\mathbf{x}$ is presented to the network, which gives the conditions in Eqs 13 and 14 for both implicit and dendritic models (notice that these two models do not differ in parameter learning). We now write the model parameters $\boldsymbol{v}$ and $W$ into block matrices:

$$\boldsymbol{v} = \begin{bmatrix} \boldsymbol{v}_k \\ \boldsymbol{v}_m \end{bmatrix}, \quad W = \begin{bmatrix} W_{kk} & W_{km} \\ W_{mk} & W_{mm} \end{bmatrix} \tag{28}$$

where $W_{pq}, p, q \in \{k, m\}$ denotes the $p \times q$ submatrices of the weight matrix $W$. Notice that for both implicit and dendritic models, the retrieval dynamics converge if and only if all error nodes become zero, that is, $\boldsymbol{\varepsilon} = \mathbf{x} - W\mathbf{x} - \boldsymbol{v} = 0$. For the dendritic model this is obvious (Eq 18), whereas for the implicit model this comes from the fact that $W$ has all its diagonal entries equal to 0 and thus cannot be an identity matrix (Eq 15). We can thus derive the following theorem on the converged retrieval dynamics of the implicit and dendritic covPCNs:

**Theorem 2** *After training, both the implicit and dendritic models retrieve the corrupted* $\mathbf{x}_k$ *given intact* $\mathbf{x}_m$ *following the dynamics specified in* Eqs 15 and 18, *which converge to the following equilibrium:*

$$W_{km}(\mathbf{x}_m - \bar{\mathbf{x}}_m) = (I_{kk} - W_{kk})(\mathbf{x}_k - \bar{\mathbf{x}}_k) \tag{29}$$

*where* $I_{kk}$ *is the* $k \times k$ *identity matrix. Using* Eq 14, *this equation is equivalent to:*

$$\hat{\mathbf{x}}_k = S_{km}S_{mm}^{-1}(\mathbf{x}_m - \bar{\mathbf{x}}_m) + \bar{\mathbf{x}}_k \tag{30}$$

The details of the derivations can be found in S1 Appendix. Notice that the equilibrium condition for implicit and dendritic covPCNs is exactly the same as that for the explicit covPCN, specified in Eq 27. This equivalence was also verified empirically in a simple 2-dimensional example shown in Fig 3D. In this example both $\mathbf{x}_k$ and $\mathbf{x}_m$ are single-dimensional scalars, and all three models retrieved the corrupted $x_2$ following the same linear equation (Eq 27). Notice that this line is also the least squares regression line, where $x_2$ is the dependent variable. It is also worth noting that this line implies that the memory "attractors" learned by the model is a line, instead of individual points, in the 2-dimensional case, which is different from the attractors learned by classical models such as Hopfield Networks [4]. In the final section of the Results, we provide an empirical investigation into this difference. Fig 3C also shows that the weight matrix $\Sigma$ of the explicit covPCN directly encodes the sample covariance matrix, whereas the other two models encode it implicitly. Due to their equivalence, the implicit and dendritic models also perform AM on the random Gaussian patterns in Fig 3A.

At this point, we have established the complete equivalence between the explicit and implicit/dendritic covPCNs, showing that the latter models can serve as perfect substitutes for the explicit model. However, in contrast to the explicit covPCN, where the learning rule (Eq 4) for the connections Σ employs non-local information, the plasticity rules (Eq 12) for the implicit and dendritic models are entirely Hebbian. Moreover, as was shown above, the neural implementation of a dendritic covPCN can be mapped to the structure of hippocampal pyramidal cells. Therefore, our proposed models are equivalent to but biologically more plausible than the explicit covPCNs.

## Model performance in AM with structured image data

In each learning iteration, the explicit covPCN needs to compute the inverse of the current weight matrix. In practice, this works well when the underlying dataset has some specific regularities, but becomes problematic when dealing with structured data, such as images of handwritten digits and natural objects. In this section, we show that the explicit model can no longer perform stable memorization and retrieval of more complex and structured data, whereas the implicit and dendritic models are able to perform AM on such data with a high level of precision, and are hence interesting from an application perspective. To do that, we replicate the pattern completion experiment performed in Fig 3, but using images sampled from the MNIST [19] and grayscale CIFAR10 [20] datasets. Fig 4A shows some retrieved examples from both datasets by all three types of single-layer covPCNs, which were trained to memorize 64 images. The visual results suggest that the explicit covPCN can no longer retrieve clear images, whereas both implicit and dendritic models can obtain visually perfect retrievals of the memories. The failure of the explicit covPCN with these structured images is due to the need to compute the inverse of Σ in each iteration (Eq 4): for low-dimensional patterns with

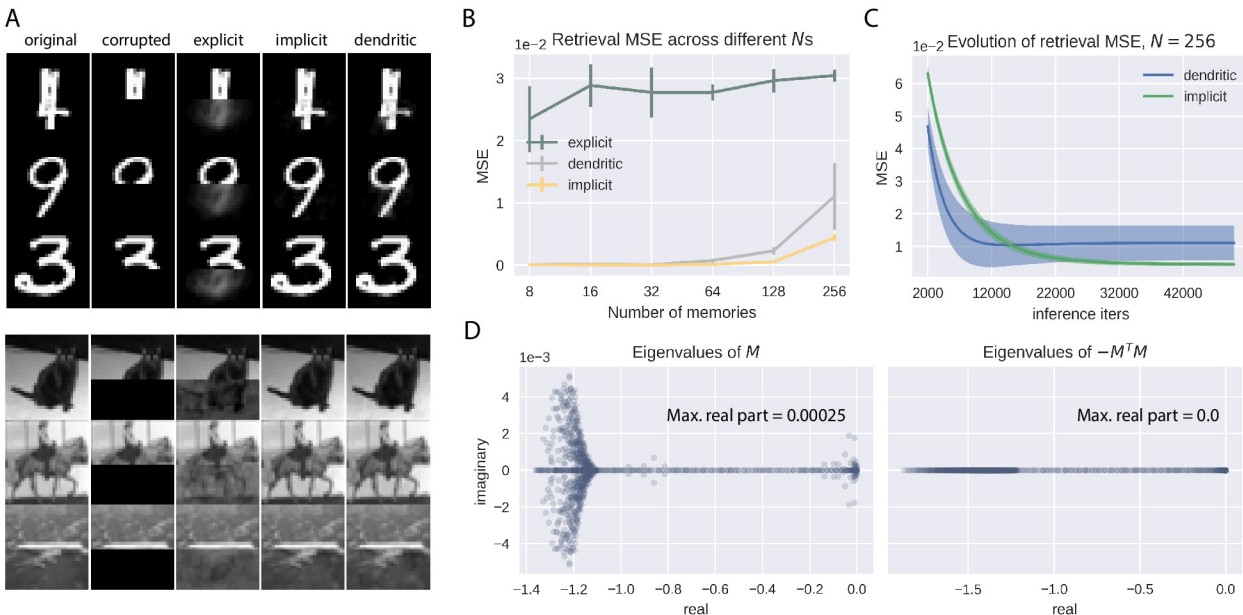

**Fig 4. Performance of the single-layer covPCNs in AM of structured images.** A: Examples of retrieved MNIST (top) [19] and grayscale CIFAR10 (bottom)[20] images by explicit, implicit and dendritic models. All models here are trained to memorize 64 images. For MNIST, the networks have $d$ = 784 neurons; for grayscale CIFAR10, $d$ = 1024. B: Retrieval mean squared errors (MSEs) of the single-layer models across multiple numbers of training memories ($N$). C: Evolution of the retrieval MSEs of the implicit and dendritic models when $N$ = 256. D: Example eigenspectra of the weight matrices defining the inferential dynamics for the dendritic (left) and implicit (right) covPCNs. Error bars obtained by 5 different seeds for image sampling. Please see main text for an explanation of the matrix $M$.

some specific regularities such as Gaussian distribution, the inverse can be precisely computed. However, when encountered with high-dimensional data such as structured images, the inverse computation can become imprecise, and the computational errors may accumulate when the model is trained iteratively, leading to blurry retrievals.

We next perform a quantitative analysis of the performance of these models, by measuring the mean squared errors (MSEs) between the original and retrieved grayscale CIFAR10 images across different numbers of images we train the models to memorize ($N$). Fig 4B shows that the explicit model indeed has much larger retrieval MSEs than the implicit and dendritic model, confirming our visual observations. The MSE gap is also larger than that in Fig 3A, where the random Gaussian patterns makes the learning of the explicit model more stable when inverting $\Sigma$. Interestingly, this plot also shows that despite the equivalence between the implicit and dendritic models when their inferential dynamics (Eqs 15 and 18) have converged, the dendritic model has larger retrieval MSEs than the implicit model, especially when $N$ is large. We investigate this phenomenon by plotting the evolution of the retrieval MSEs of both implicit and dendritic models during inference, when $N = 256$. As can be seen in Fig 4C, the MSE of the dendritic model fails to decrease to the same (lower) value as that of the implicit model, and it also fails to converge. To understand the distinction between the implicit and the dendritic covPCNs in inference, recall that their inferential dynamics are described by two linear differential equations (Eqs 15 and 18). We now define:

$$M = W - I \qquad (31)$$

where $W$ is the learned weight matrix in the implicit/dendritic models (note that the learning of these two models are identical) and $I$ is the identity matrix of the same size as $W$. We can therefore rewrite the inferential dynamics as:

$$\Delta x_{\mathrm{im}} = \beta(-M^T M x), \quad \Delta x_{\mathrm{den}} = \beta(Mx) \qquad (32)$$

where "im" stands for implicit, and "den" stands for dendritic. Notice that in general, the stability of a linear dynamical system $\Delta x = Ax$ for some matrix $A$ is determined by the eigenvalues of $A$: the system is stable if and only if the real part of all eigenvalues of $A$ are non-positive [23]. As we show in Fig 4D, the largest eigenvalue of $-M^T M$ is 0, suggesting stable inferential dynamics for the implicit covPCN. This is also generally true as $-M^T M$ defines a negative (semi-)definite matrix for any $M \neq 0$, which has all eigenvalues smaller than or equal to 0. On the other hand, Fig 4D shows that the largest eigenvalue of $M$ is greater than 0, indicating unstable dynamics for the dendritic model. Thus, although the implicit and dendritic models have the same fixed point of their dynamics during retrieval, the neural dynamics defined by the dendritic model may become unstable, preventing itself from the convergence. For the dendritic model there are no theoretical reasons that guarantee negative eigenvalues of $M$, so they may become positive as the values in that matrix grow with learning more patterns. The experiments with MNIST [19] yielded similar results, and we describe them in S1 Fig.

## Performance of hybrid models with natural images

While performing well on small subsets of structured data, the purely recurrent implicit/dendritic models present some problems: first, the recurrent structure only takes account of the hippocampal structure, but does not correctly represent the hierarchical structure that connects it to the neocortex; second, the number of parameters is quadratic to the dimension of the inputs, which does not allow it to work on high-dimensional images, such as high quality images. For example, consider a recurrent network trained on $224 \times 224$ pictures of the ImageNet dataset [24]. Every image consists of $\sim 150000$ pixels, and hence needs a network with

the same number of neurons. A recurrent network of this size would be impossible to train without an exceptional amount of computational power, as it has $\sim 40$ billions of parameters. On the other hand, a hierarchical structure that precedes the hidden implicit layer guarantees the flexibility of choosing the number of parameters. For example, a network with 7 hidden layers, followed by a recurrent implicit layer, all of dimension 1024, would be more than 200 times smaller than the above implicit model for ImageNet, and hence feasible to be trained. In fact, we have successfully trained this model on high-dimensional ImageNet pictures. Particularly, we trained this 7-layer network with 100 samples from the ImageNet dataset. We then defined successful retrievals as those with retrieval MSEs smaller than $5e^{-3}$, and found that when presenting the network a partial cue consisting of 1/2, 1/4 and 1/8 of the original pixels, the model has successfully recovered, respectively and on average, 100, 97, and 44 of the original memories. Similar performance was also obtained with the colored CIFAR10 dataset [20] and examples of colored CIFAR10 images can be found in S2 Fig.

We then examine whether the hybrid structure leads to better retrieval performances compared to the single-layer models. To do that, we trained an implicit covPCN and a hybrid model with a topmost implicit covPCN to memorize the same subsets of grayscale CIFAR10 images (1024 pixels in total) of varying number of images $N$, and initialized the retrieval with half-covered partial cues. To compare with earlier works [12] fairly, we also trained a purely hierarchical PCN to perform the same AM task. The sizes of the networks are chosen such that they have approximately the same number of parameters. The hybrid model we use has 3 layers: the sensory layer has 1024 neurons corresponding to the pixel space, with 512 neurons in the hidden layer, and 512 neurons in the topmost implicit layer. We construct the hierarchical model by replacing the topmost implicit recurrent layer with 2 hierarchical layers of size 512. We chose such a configuration of hidden sizes and number of layers because in general, an implicit covPCN with $d$ neurons (for $d$-dimensional inputs) will have $d(d-1)$ parameters. A hybrid model with $d$ sensory neurons, one feedforward hidden layer of size $d/2$ and a topmost implicit layer of size $d/2$ will have approximately the same number of parameters. Replacing the final implicit layer with two feedforward layers of the same size results in a hierarchical PCN with roughly the same number of parameters. This ensures the fairness of comparison across models, and is illustrated in Fig 5A, where we also included the number of neurons in each layer used in our experiments next to each layer, as well as the number of connections at the top. As Fig 5B shows, the performance of the implicit model is in fact slightly better than the hybrid implicit model and the purely hierarchical model when they have the same number of parameters. Indeed, the retrieval MSE of the implicit model should be the smallest among all linear retrieval functions: notice that the retrieval $\hat{\mathbf{x}}_k$ of the implicit model defined in Eq 27 is the least squares solution if we regress $\mathbf{x}_k$ on $\mathbf{x}_m$. Although the exact retrieval function of the hybrid and hierarchical models is unclear due to their nonlinearity, it has been shown that a two-layer linear PCN with 1 neuron in the hidden layer yields linear retrievals following the principal components of the sensory data [15], shedding light on the slightly worse performance of the multi-layer models. However, we note that the similar performance in retrieval fidelity between the implicit and hybrid models does not make the hierarchical structure in the hybrid model redundant, as the functionality of the brain is far more heterogeneous than merely memorization, and many functions require the hierarchical structure. For example, the hierarchical structure can support the learning of meaningful representations of the sensory inputs, which can be utilized to perform other tasks such as classification, while paying only a negligible cost in memory retrieval tasks.

We further compared the implicit and dendritic models when they are "plugged in" as the topmost recurrent layer in the hybrid model. Fig 5B shows that the hybrid dendritic model performs identically to the hybrid implicit model, contrasting their performance difference in

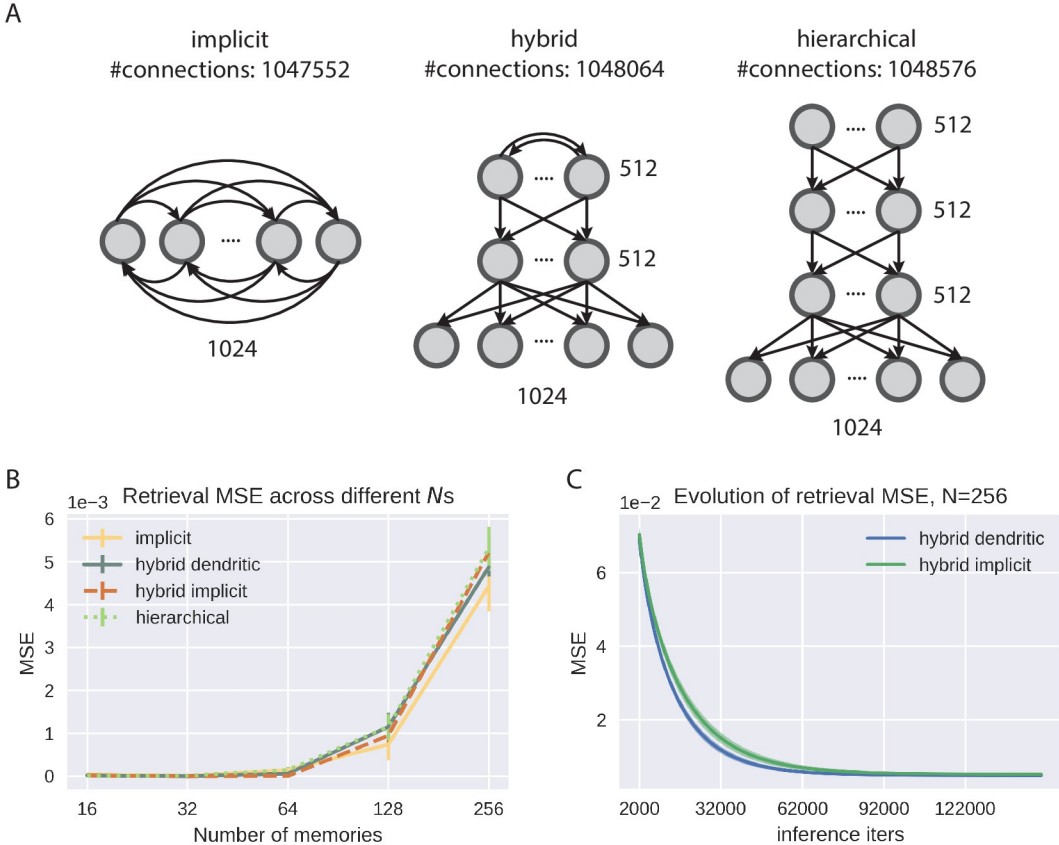

**Fig 5. Performance of the multi-layer models.** A: Demonstration of how we keep the number of parameters across different models to be roughly the same. B: Retrieval mean squared errors (MSEs) of the multi-layer models across multiple numbers of training memories (*N*). The curve for the implicit model is the same as the one in Fig 4. C: Evolution of the retrieval MSEs of the implicit and dendritic hybrid models when *N* = 256. Error bars obtained by 5 different seeds for image sampling.

the single-layer case. During the inferential iterations, the retrieval MSEs of both models also converge stably to the same value (Fig 5C). In other words, the introduction of the neocortical layers remedies the unstable inferential dynamics due to the biologically more plausible dendritic structure in our models. We suspect that the hierarchical pre-processing of the sensory inputs regularize the covariance matrix, such that the topmost recurrent weight matrix *W* defines stable inferential dynamics for the dendritic layer. However, since the top-layer inference is no longer a simple linear dynamical system (Eq 22), the connections between a regularized *W* and the stability of the dynamics may not be straightforward.

## Nonlinear covPCNs learn individual attractors

The implicit covPCN we have investigated so far assumes that the relationship between neurons in the recurrent network is linear, i.e., projections into neuron $i$ is $\sum_{j \neq i} W_{ij} x_j$. Here, we introduce an ad hoc nonlinearity into the definition of the free energy—as motivated by previous heuristics in neural networks and machine learning:

$$\mathcal{F} = -\frac{1}{2} \| \mathbf{x} - W f(\mathbf{x}) - v \|_2^2 \tag{33}$$

where $f(\cdot)$ is a nonlinear function. One way to understand these heuristics more formally is to

appreciate that the free energy is a statement about the underlying generative model that generates inputs (e.g., images) from some latent causes. Although linear covPCNs have provided nice analytical results discussed above, nonlinearities can speak to more expressive generative models similar to the hierarchical and hybrid schemes demonstrated above. The plasticity rules of this model will be the same as Eqs 11 and 12, with the error $\boldsymbol{\varepsilon}$ changing to $\mathbf{x} - Wf(\mathbf{x}) - \boldsymbol{v}$. The neural dynamics during inference follows:

$$\Delta\mathbf{x} = \beta\frac{\partial\mathcal{F}}{\partial\mathbf{x}} = \beta(-\boldsymbol{\varepsilon} + f'(\mathbf{x}) \odot W^T\boldsymbol{\varepsilon}) \tag{34}$$

We first train this nonlinear implicit model to perform the same AM task as before, where the model is trained to memorize different numbers of grayscale CIFAR10 images, and then retrieve these memories from images with covered bottom half. For these experiments with CIFAR10 we used $tanh()$ as the nonlinear function $f()$. Fig 6A shows that the performance of the nonlinear implicit covPCN is worse than that of the linear model in this completion task, especially when $N$ is large. To examine the reasons for this performance difference, we then

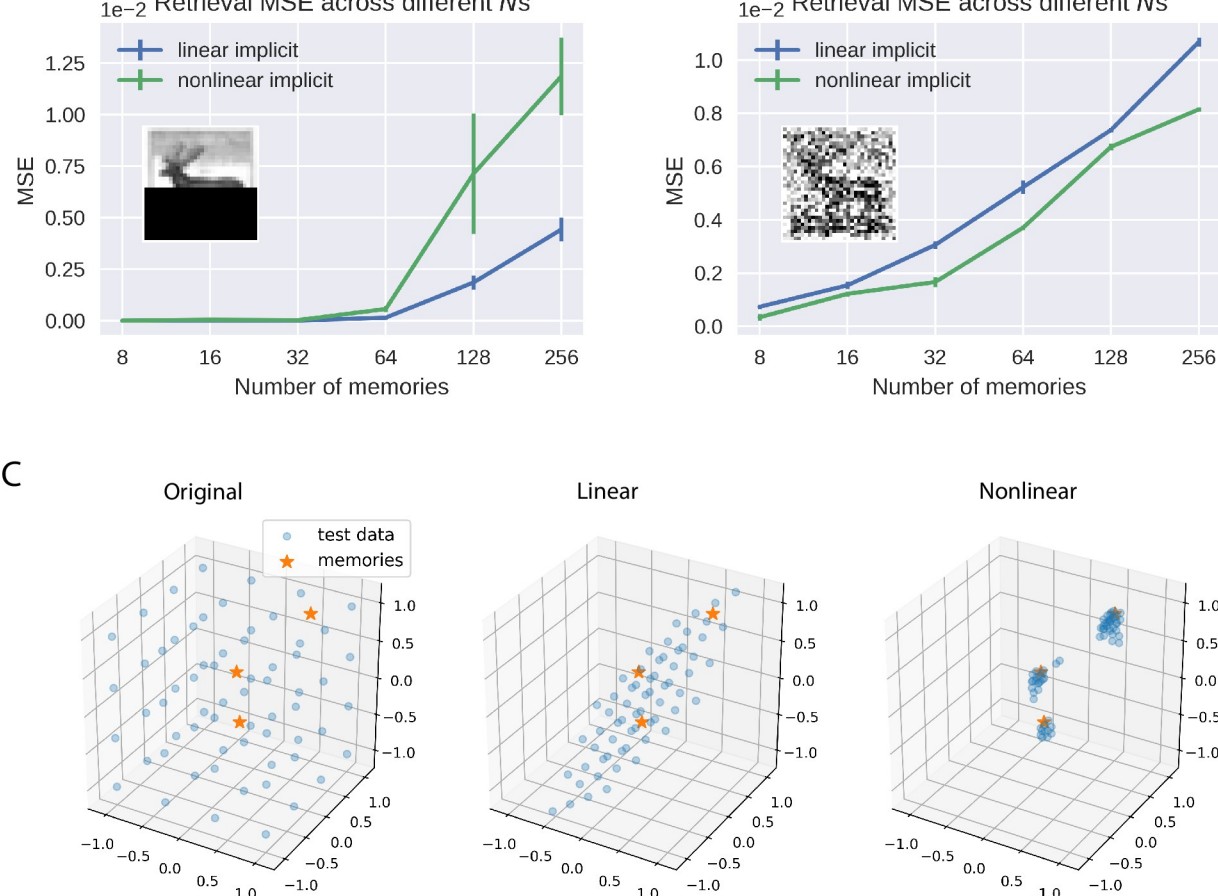

**Fig 6. Comparison of linear and nonlinear implicit covPCN.** A: Performance of linear and nonlinear implicit models in the completion task with varying $N$s. B: Same as A, but with the denoising task, where cues are memories with Gaussian noise of variance 0.1. C: A simple 3-dimensional example, where stars are data points the networks were trained to memorize. After training we ran inference on both linear and nonlinear models, initialized with grid test data drawn from the range $[-1, 1]^3$. The position of the test data at convergence of inference indicates the shape of attractors. Images taken from the CIFAR10 dataset [20].

corrupt the memories differently by adding Gaussian noise with variance 0.1, and initialize the retrieval dynamics with these noisy memories. Fig 6B shows that with this denoising task, the linear and nonlinear implicit covPCNs perform similarly, although the retrieval MSE of the nonlinear model is overall lower. We hypothesize that the difference across different retrieval tasks results from the attractor structures learned by the linear and nonlinear models. Classical nonlinear models, such as the Hopfield Networks [4], usually learn individual attractors corresponding to each memorized item. On the other hand, as we have demonstrated in Fig 3D, the linear implicit covPCN appears to learn a line attractor in the 2-dimensional space. We generalize this observation by training both linear and nonlinear models to memorize 3 random data points in range $[-1, 1]$ in $\mathbb{R}^3$. After training, we initialize the inference with test data in a grid from the range $[-1, 1]$, and examine where these test data points are attracted to at the convergence of the inferential dynamics. Fig 6C shows that the attractor formed by the linear model is a hyperplane, where any point on this plane defines the lowest energy. On the other hand, the test data points all converge to the memories learned by the nonlinear model, suggesting that the nonlinear model indeed learns individual attractors.

The observation above in turn helps explain the performance difference in completion and denoising tasks between the linear and nonlinear models. Notice that the completion task is inherently more challenging than the denoising task in terms of retrieval, as the corrupted pixels contain no information about the original images. Therefore, in the completion task, individual attractors formed by the nonlinear model will confound with each other more severely than in the denoising task, so that more retrievals will converge into the wrong attractors. The retrieval MSE of the nonlinear model thus comes from these incorrect attractor choices. With the linear model, the retrieved data points are always pulled towards the least squares regression hyperplane, such as the regression line in Fig 3. The MSE of the linear model therefore only comes from the least squares prediction, which is the minimum across all linear solutions. Thus, when we average across the whole dataset, a relatively large number of totally wrong retrievals will produce a higher MSE than the retrievals from the least squares predictions. On the other hand, in the less challenging denoising task, the confounding effect between attractors of the nonlinear model will be less severe, yielding a smaller retrieval MSE across the whole dataset than in the completion task. Moreover, the least squares prediction from the linear model is less reliable in the denoising task, as all entries of the data are corrupted, i.e., the independent variables in the regression problem are also changing. Therefore the retrieval MSE of the linear model will increase, resulting in the observation in Fig 6A and 6B.

## Discussion

### Summary

This work introduces a family of covPCNs for AM tasks, providing a possible mechanism of how the recurrent hippocampal network may perform AM via predictive processing, which also unifies two disparate modelling approaches to the hippocampal network, namely covariance learning and PC. First, we identified that a previously proposed PCN already considered the learning of parameters encoding covariance in an explicit manner [16, 17], but was never tested for AM tasks. We showed that, both theoretically and empirically, this model is able to perform AM on random patterns. However, this *explicit covPCN* is neither biologically plausible nor numerically stable, due to the inverse term in its learning rule. We address both limitations by proposing a model we call *implicit covPCN*, which also learns the covariance matrix, but in an implicit manner. More importantly, it adopts a Hebbian learning rule that is biologically more plausible than the learning rule of the explicit covPCN. We then showed that this model can be further simplified by introducing an apical dendritic architecture, which is even

**Table 1. Summary of covPCNs discussed in this work.**

|  | Hebbian | Parameter flexibility | Random pattern | Structured pattern |
|---|---|---|---|---|
| explicit | ✗ | ✗ | ✓ | ✗ |
| implicit | ✓ | ✗ | ✓ | ✓ |
| dendritic | ✓ | ✗ | ✓ | ✓ |
| hybrid | ✓ | ✓ | ✓ | ✓ |

"Hebbian" denotes whether the model employs the Hebbian learning rule; "Parameter flexibility" denotes whether we can freely control the number of parameters of the network, i.e., whether the number of parameters in the model is not constrained to scale quadratically with the input size; "Random/Strctured pattern" denotes whether the model can perform AM on random/strctured patterns.

more biologically realistic. We named it the *dendritic covPCN*. In theory, we showed that both the implicit and the dendritic models are exactly equivalent to the explicit covPCN in AM tasks if their inference converges, although the dendritic model may suffer unstable inferential dynamics. Nevertheless, in practice, both implicit and dendritic models can be trained to memorize complex patterns such as handwritten digits and natural images. However, these models are computationally and memory expensive, a drawback that limits its applicability when confronted with more complex datasets. We hence solve this problem by combining our implicit covPCN with a hierarchical PCN [11, 12], to model the predictive interaction between the hippocampus and neocortex [9]. We showed that this hybrid network can memorize and retrieve large-scale images in a parameter-flexible manner. We further conducted empirical analysis of the memory attractors learned by our linear implicit model, and showed that the memorized patterns are all stored on a hyperplane attractor of the inputs space. We found that nonlinearities are needed to obtain individual point attractors corresponding to each memorized pattern, similar to those observed in Hopfield Networks [4]. Table 1 below summarizes the property of each model discussed in this work. Overall, our models benefit the theoretical understanding of the computational principles adopted by the hippocampal network, by providing a unitary account for two disparate theories of how the hippocampus performs AM tasks.

## Relationship to other models

**Predictive coding networks.** Our proposed models form a subset of the PC models [11, 16, 17], characterized by neural architectures (e.g., neurons and dendrites) encoding the mismatch between internally generated predictions and sensory inputs, i.e., error signals. Linear PC was first introduced for the compression of timeseries data in the 1950s [25]. In the neurosciences, it was originally proposed to explain the efficient processing of visual inputs to the retina [26] and subsequently used to model hierarchical processing in the visual cortex, with a special focus on extra-classical receptive field effects [11]. As noted above, PC is a particular class of variational inversion schemes for hierarchical generative models under Gaussian or parametric assumptions [27]. Some versions of PC deal with state space models and implicit predictions about dynamics [28]; it was later extended to account for a range of neuronal responses from various brain regions such as the retina and auditory cortex [29]. The versatility of the PC framework enabled it to be later adopted as a general, high-level model for representation learning in the brain [16, 17], i.e., models for describing how sensory inputs are represented in the cortex. The explicit covPCNs described in this work are essentially the PC models with parameters encoding the precision (covariance) matrix [16–18] with a single layer architecture. These models introduced the covariance matrix to represent the uncertainty of different input features, by differentially weighting error signals derived from different input

sources. However, to our knowledge they have never been tested in AM tasks. Earlier works have also identified the implausibility of the learning rule for the covariance parameters, and addressed this issue by introducing additional inhibitory interneurons on top of error neurons [18]. On the other hand, our implicit/dendritic covPCNs naturally circumvents the implausible learning rule by introducing a new set of recurrent weights $W$. Furthermore, our nonlinear covPCN is reminiscent of the sparse coding schemes of the kind found in independent component analysis [30]. This suggests an interesting relationship between sparse coding, PC and the properties evinced by the numerical studies above.

In recent years, multiple lines of research have revealed the potential of PC in modelling various brain functions and in performing complex machine learning tasks. Most notably, the exploration of purely hierarchical PCNs in AM tasks by Salvatori et al. [12] has directly inspired our work, and their model was also employed in our work to construct the hybrid PCN. Recurrent PCNs for AM tasks have also been proposed recently [31]. However, the covariance-learning property of these recurrent PCNs, as well as their connections to earlier PCNs with precision matrices [16–18], were not fully investigated in these works. More broadly, the approximation of PC to backpropagation [32], the most commonly used learning rule of modern artificial neural networks, has been extensively investigated and discussed in recent works [15, 33–37]. The PC algorithm has also been scaled to adapt to various modern machine learning architectures [31, 38–41] and studies have also shown that it can approximate backpropagation in arbitrary computational graphs [42]. These works suggest that it is possible to implement PC within modern deep neural networks, making our PC-based AM networks potential candidates of efficient memory-storage machines in artificial intelligence.

**Models for associative memory and the hippocampus.** The covariance-learning property of our proposed models relates them directly to earlier AM models [4, 43–45] that memorize random patterns by learning the covariance or correlation between features, as well as their modern variations for complex patterns [46–48]. The simplicity of these models has enabled significant developments on the understanding of their theoretical properties, especially the memory capacity [49–51]. Thus, the connection of our covPCNs to these classical AM models makes the capacity computation of PC-based AM models an intriguing future direction.

Independent from these earlier models, new theories and models for AM have also been proposed recently, such as those based on the autoencoder [52]. Although these models work well in performing AM retrieval tasks, they usually lack resemblance to the known anatomy of the hippocampus, and are trained with biologically implausible learning algorithms such as backpropagation [32]. On the other hand, PC-based models for AM [12] provide a biologically more plausible approach that follows both local, Hebbian learning as well as the predictive nature of the hippocampus [8, 9]. Our models take a step further by introducing recurrent connections to PCNs, which capture the connectivity pattern of the hippocampal network.

Importantly, it is worth noting that the comparison between PC-based hierarchical memory networks [12] and earlier models for AM such as the Hopfield Networks [4] has been thoroughly investigated by Salvatori et al. [12]. In particular, they found that classical Hopfield Networks [4] and their modern variations [46] performed much worse than the hierarchical PCN in memory retrieval tasks. The retrievals with Hopfield Networks can also easily collapse into one memorized item, resulting in large retrieval errors. Since we have shown a similar performance of our models in AM tasks to the hierarchical PCNs, one can infer how our covPCNs compare to the classical models.

The aforementioned works focused mainly on computational models for associative memory, whereas another track of research concerns the modelling of the entire hippocampal

network with sub-region connectivities, including the entorhinal cortex (EC), dentate gyrus (DG), CA3 and CA1 areas. For example, both Treves and Rolls (1992) [6] and O'Reilly and McClelland (1994) [5] proposed models containing the sparse DG region for pattern separation, and the recurrent CA3 block that performs pattern completion through AM. The characteristic connectivity pattern between these regions, such as the mossy fiber pathway and the perforant pathways, were also considered in these models. More recent models of these sub-regions employ modern machine learning architectures to study place cells in the hippocampus [53]. Our single-layer covPCNs can be viewed as models of the single recurrent CA3 area performing AM, or a full graph of the hippocampal network where sub-region specifications are temporarily ignored but can be achieved by modifying graph topology of this network via PC, following the mechanism proposed by Salvatori et al. (2022) [31]. Furthermore, our hybrid PCN attempted to model the hippocampo-neocortical interaction using a unified computational principle, i.e., PC. This relates them to earlier computational models for the mediation of stimulus representations in the cortex by hippocampus after a memory is recalled [54], as well as computational models of the hippocampo-neocortical system that helped explain the memory consolidation phenomenon [55].

**Models with dendritic computation.**   In the dendritic covPCN proposed in this work we computed error signals within apical dendrites, rather than within explicit error nodes in original PCNs. This relates our model to works investigating dendritic computation. Earlier works have shown that it is possible to construct multilayer neural networks with dendritic compartments encoding errors, which also approximates backpropagation [21]. It was later illustrated by Whittington and Bogacz that these models are closely related to PC [56]. PC with dendritic computations has also paved the way for spiking neural networks [22], pushing it towards more biological plausibility. Our dendritic covPCN differs from these dendritic models, in that the dendritic formulation results naturally from a stop-gradient operation, rather than an artificial construction of dendrites. Broadly speaking, recent studies have shown that incorporating dendritic architectures into artificial neural network may benefit transfer and continual learning, and help the design of neuromorphic hardware with lower energy consumption (see [57, 58] for reviews of dendritic computation in artificial neural networks), suggesting other potential advantages of dendritic PCNs beyond biological plausibility.

## Relationship to experimental data

Our proposed models all belong to the family of PCNs, which assumes that computations in the cortex are carried out by generating predictions of activities representing sensory inputs. In general, it has been shown that this assumption is consistent with the known anatomy of the cortex, and that several cortical microcircuits can implement the PC algorithm [59–62].

More specifically, experimental evidences have suggested mechanisms of predictive processing within the hippocampus and related areas. It has long been postulated that the hippocampus predicts upcoming sensory inputs, in both spatial and non-spatial experimental setups [7, 8, 63, 64]. The special feedforward circuitry of the hippocampus may also be an evidence for predictive processing: Lisman [65] suggested that the CA1 region of the hippocampus serves as a "mismatch" detector, encoding the error between sensory reality and the internally generated prediction based on past events communicated through the feedforward pathway from CA3 to CA1. However, although hippocampal cell populations that fire exclusively following erroneous trials in associative learning tasks have been found in animals [66, 67], it is unclear, at the single-cell level, whether prediction error representations exist in the hippocampus, and in what form they exist: are there neurons encoding them, or are they encoded in specific neuronal structures such as apical dendrites? Our models inform future experimental

works to answer these questions by making the prediction that hippocampal neurons can representation prediction errors, via somatic or dendritic activities.

It is also worth pointing our that the experiments in our work focused mainly on "spatial" predictions e.g., predicting each pixel using other pixel values. Although predictions made through different pixels or patches of a static image can be interpreted as eye saccade across a large visual scene, prediction of sequential sensory inputs is more realistic and consistent with experiments mentioned above. Modelling temporal PC models while retaining the plausible implementation discussed in this work will be a focus of our future explorations.

### Future directions

Following the discussions above, there are several directions in which this work should develop.The theory needs to be extended to modelling sub-regions of the hippocampus using PC. Rather than a simple, fully-connected recurrent circuit, the hippocampal network contains multiple sub-regions with different possible functionalities. For example, the DG area is postulated to perform pattern separation through its sparse activities, while the CA3 is believed to be the associative area and CA1 acts as a decoder and mismatch detector [5, 6, 65]. Moreover, the connectivity between these regions also has a unique pattern. For example, the EC-DG-CA3 connection is fully feedforward and unidirectional, whereas CA3 also receives direction input from EC.It will be important to investigate how these other regions could be included in the PC framework. Notably, we have shown in this work that a simple stop-gradient operation in PCNs could yield unidirectional connections in a recurrent network, which provides a potential approach of modelling the hippocampal sub-region connectivity.

Another direction in which the work can develop is temporal prediction in the hippocampus. While abundant experimental evidences have shown the predictive nature of the hippocampus, the observed predictions are mainly sequential or temporal, i.e., predicting sensory input in the future. This temporal property of the hippocampal prediction is not reflected in our PC models. Earlier and recent works have attempted to model cortical temporal prediction using a Kalman filter [10, 28], where the hierarchical prediction of sensory inputs made by the hidden state can potentially be implemented within the hierarchical PCNs [11]. Another track of research assumes that both sensory input and neural response dynamics along the temporal dimension are contained by attractor manifolds, but on different levels [68, 69]. Thus, higher-level attractor of neural responses can generate controls over lower-level attractors of responses or sensory input streams, which can be described within the hierarchical PC/free energy framework [11, 16]. However, it remains an open question whether these temporal PC models can be implemented within simple and plausible neural circuits similar to those discussed in this work. It is also possible to combine this direction of research with sub-regions of the hippocampus, as the CA1 area was postulated to encode the mismatch between inputs from CA3 and those from EC [65], whereas recent experimental works have found that the EC-CA1 inputs is prioritized over the CA3-CA1 inputs when a new sensory stimulus is presented, suggesting that the mismatch encoded in CA1 results from a temporal delay [70]. These findings shed light on a hippocampal sub-region approach to temporal PC.

Moreover, in our current work, we have focused on learning covariance or precision matrices inherent in a batch of training images, which can be implemented within a single-layer network. On the other hand, hierarchical implementations of PC often use more expressive generative models, in which the covariance or precisions at various hierarchical levels can change over time. This means that instead of learning the covariances, they are inferred, leading to the notion of dynamic precision weighting of prediction errors [71, 72]. This is also particularly interesting from a machine learning point of view, because it provides a model of

attention from a neuroscience perspective—that may be related to the kinds of attention found in transformers and the like [73].

## Supporting information

**S1 Appendix. Expression for retrieved patterns in the covPCNs (Eq 27).**
(PDF)

**S1 Fig. Experimental results with MNIST [19].** A: Retrieval MSEs of the single-layer models across multiple $N$s. B: Retrieval MSEs of the multi-layer models across multiple $N$s.
(PDF)

**S2 Fig. Examples of retrieved color CIFAR10 images [20] with different levels of corruptions.** We also performed some preliminary experiments with the dendritic model on colored CIFAR10 images, and show some examples here. Particularly, we found that when presenting the network a cue consisting of 1/2 (panel A), 1/4 (panel B) and 1/8 (panel C) of the original pixels, the model has successfully recovered, respectively and on average, 89, 79, and 49 of the original memories in the experiments using colored CIFAR10.
(PDF)

**S1 Table. Implementation details of experiments with grayscale CIFAR10.** For all sample size of memories $N$, we use a batch size of $N/8$. For the inference iterations with the multi-layer models, the first number 400 is the number of inference iterations during training and within each training iteration. The second number 100000 is the number of inference iterations used to retrieve the original patterns.
(PDF)

**S2 Table. Implementation details of experiments with MNIST.** For all sample sizes of memories $N$, we use a batch size of $N/8$. For the inference iterations with the multi-layer models, the first number 400 is the number of inference iterations during training and within each training iteration. The second number 100000 is the number of inference iterations used to retrieve the original patterns.
(PDF)

## Author Contributions

**Conceptualization:** Rafal Bogacz.

**Formal analysis:** Mufeng Tang.

**Investigation:** Mufeng Tang, Tommaso Salvatori.

**Software:** Mufeng Tang, Tommaso Salvatori, Yuhang Song.

**Writing – original draft:** Mufeng Tang.

**Writing – review & editing:** Tommaso Salvatori, Beren Millidge, Yuhang Song, Thomas Lukasiewicz, Rafal Bogacz.

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
