## [Decision Letter · Decision Letter 0]

12 Dec 2022

Dear Dr. Bogacz,

Thank you very much for submitting your manuscript "Recurrent predictive coding models for associative memory employing covariance learning" for consideration at PLOS Computational Biology.

As with all papers reviewed by the journal, your manuscript was reviewed by members of the editorial board and by several independent reviewers. In light of the reviews (below this email), we would like to invite the resubmission of a significantly-revised version that takes into account the reviewers' comments.

We cannot make any decision about publication until we have seen the revised manuscript and your response to the reviewers' comments. Your revised manuscript is also likely to be sent to reviewers for further evaluation.

Sincerely,

Xuexin Wei

Academic Editor

PLOS Computational Biology

Daniele Marinazzo

Section Editor

PLOS Computational Biology

Reviewer's Responses to Questions

**Comments to the Authors:**

Reviewer #1: I enjoyed reading this compelling and thoughtful treatment of different predictive coding algorithms — and how they may be implemented by the hippocampus. I thought your explanation of this sort of predictive coding (and the underlying schemes) was clear and accessible. However, I think you need to make it easier for the reader to understand why your scheme works. Furthermore, it might help both you and the reader understand the various implementations of PCN you have considered in the larger context of predictive coding in neuroscience and engineering. I have suggested how you could do this by unpacking and qualifying some of your key points.

Major points

It would be helpful for the reader to explain why your simplifications work — and take the opportunity to explain what kind of predictive coding you are using to illustrate your ideas. I would recommend the following at the end of the introduction:

“In summary, we describe a series of simpler and more biologically plausible implementations of predictive coding that we frame in terms of associative memory and hippocampal function in the brain. To unpack the basic ideas, we will focus on a simple kind of predictive coding, in which we ignore dynamics and temporal prediction. Furthermore, we will restrict our initial analyses to linear systems, under standard parametric assumptions (i.e., Gaussian assumptions about random effects). In this setting, a memory is simply the ability of the PCN to recognize the most likely cause of a particular pattern of inputs (e.g., an image). We are not addressing episodic memory – of the sort sometimes associated with hippocampal function — rather, we are focusing on how statistical regularities in a series of inputs are learned and then used to predict the underlying cause of a corrupted or random input.

In terms of inference and learning, predictive coding can be regarded as a special class of optimization schemes — that maximize the marginal likelihood of generative models via the minimization of variational free energy — under Gaussian or parametric assumptions. These Gaussian assumptions mean that the requisite free energy — gradients that drive inference and learning — can be expressed as prediction errors, which is the definitive aspect of predictive coding. The key thing that we bring to the table is a reparameterization of the prediction errors, which leads to simplified forms of a free energy. Crucially, the simplified forms of free energy we consider all share the same minima or fixed points; thereby leading to the same inference and learning. Practically, this allows us to drop certain terms from the gradients (i.e., prediction errors), leading to more robust convergence to free energy minima and, crucially, affording more biologically plausible implementation."

When you first mention the energy or objective function on line 100, please refer explicitly to “(variational) free energy” (as an upper bound on marginal likelihood). Your work has moved beyond the use of arbitrary energy functions found in neural networks and machine learning and is using an energy which bounds model evidence or marginal likelihood.

On line 131, please include a footnote or sentence along the following lines:

“Although we refer to these estimates maximum likelihood estimates (MLE), in more general formulations they would correspond to maximum a posteriori (MAP) estimates. However, because we have not placed any prior constraints on the implicit generative model implied by equation (1), the MLE and MAP become the same thing."

The section on nonlinear predictive coding networks and the learning of individual attractors, is interesting but could be removed. I suggest this because the particular functional form of the nonlinearity in (33) is not motivated in any principled way. And the connection with attractors needs to be developed more rigorously. In other words, I think you need to think about this more carefully before committing to the literature.

Alternatively, you could qualify this section by including the following at the end:

“In (33) we have introduced an ad hoc nonlinearity into the definition of the free energy – as motivated by previous heuristics in neural networks and machine learning. One way to understand these heuristics more formally is to appreciate that the free energy is a statement about the underlying generative model that generates inputs (e.g., images) from some latent causes. For example, the generative model implied by the free energy in equation (1) is a general linear model, which explains why we can understand its inversion with PC in terms of regression coefficients and the standard linear algebra found in parametric statistics.

Adding nonlinearities to generative models of this sort usually leads to sparse coding schemes of the kind found in independent component analysis (Olshausen and Field, 1996). This suggests an interesting relationship between sparse coding, predictive coding and the properties evinced by the numerical studies above. Clearly, nonlinearities also speak to more expressive generative models of the kind with we can leverage in the hierarchical and hybrid schemes demonstrated above. In future work, we will revisit the above schemes starting from the underlying generative model, from which the free energy functional and their gradients (i.e., prediction errors) can then be derived."

On line 582, you provide a brief history of PC. This history makes it appear that you do not know the predictive coding literature or developments outside machine learning. I recommend you speed read: Does predictive coding have a future? | Nature Neuroscience. And include the following to summarize PC:

“Linear predictive coding was first introduced for the compression of timeseries data in the 1950s (Elias, 1955). In the neurosciences, it was originally proposed to explain the efficient processing of visual inputs to the retina (Srinivasan et al., 1982) and subsequently used to model hierarchical processing in the visual cortex, with a special focus on extra-classical receptive field effects (Rao and Ballard, 1999). As noted above, predictive coding is a particular class of variational inversion schemes for hierarchical generative models under Gaussian or parametric assumptions (Friston, 2008). Under state space generative models, the corresponding scheme is known as a Kalman filter for linear models – and an extended Kalman filter for non-linear models (Rao, 1999). Usually, predictive coding deals with state space models and implicit predictions about dynamics it was later extended to account for a range of neuronal responses from…"

In light of the above, you might want to refer to the subsequent developments of predictive coding based upon the variational formulation. The work based upon references [23] and [24] means that your comment on line 720 may need to be revised. In other words, there is quite a large literature on predictive coding and temporal prediction and learning. For example: (Friston and Kiebel, 2009; Kiebel et al., 2009). There are some entertaining examples using birdsong; especially numerical illustrations of predictive coding that involve learning – or indeed recognizing who is singing.

You may also want to include the following in your discussion of future directions:

“In our current work, we have focused on learning covariance or precision matrices inherent in a batch of training images. Hierarchical implementations of predictive coding often use more expressive generative models, in which the covariance or precisions at various hierarchical levels can change over time. This means that instead of learning the covariances, they are inferred; leading to the notion of dynamic precision weighting of prediction errors (Clark, 2013; Limanowski, 2017). This may be an interesting avenue to pursue; particularly, because it provides a model of attention from a neuroscience perspective – that may be related to the kinds of attention found in transformers and the like."

Minor points

Line 47, you might replace “dichotomy” with “dialectic"?

Line 144, I did not quite understand the phrase "corresponding training data point" perhaps it would be clearer to say:

“For example, if x~ is a corrupted or noisy data point, the equation above will drive it to its nearest point within the learned Gaussian density. One can see this from equation (5) in which the estimate is pulled towards the mean in proportion to the precision or inverse variance."

Line 166: did you mean to say: "which are not generated by sampling from a Gaussian distribution"?

Line 268: please say "as a memory index".

Line 286: please say "a property never previously demonstrated with this model. We then derive an expression for a retrieved…"

For each figure (when appropriate) could you specify the number of latent states (i.e., d) used for the simulations. For example, you clearly used two dimensional causes for Figure 3. However, it is not clear how many states (i.e., the dimensionality) you used for the hierarchical and hybrid schemes. It would be easiest for the reader to have this information in each figure legend (as opposed to a table).

Line 360, please replace “encode” with "encodes"

In the legend to figure 4, please add “(please see main text for an explanation of the matrix M)”.

Line 474, please replace "functionalities" with "functionality"

Line 475: please replace "requires" with "require".

Line 498: please replace "negative energy function with "negative free energy function".

Line 683: You might want to refer to the work of Jeffrey Gray, who spent much of his life theorising about prediction errors and the role of the hippocampus in learning (for example, learning and latent inhibition).

I hope that these suggestions help should any revision be required.

Clark, A., 2013. The many faces of precision (Replies to commentaries on "Whatever next? Neural prediction, situated agents, and the future of cognitive science"). Front Psychol 4, 270.

Elias, P., 1955. Predictive coding–I. IRE Transactions on Information Theory 1, 16–24.

Friston, K., 2008. Hierarchical models in the brain. PLoS Comput Biol 4, e1000211.

Friston, K., Kiebel, S., 2009. Predictive coding under the free-energy principle. Philosophical transactions of the Royal Society of London. Series B, Biological sciences 364, 1211-1221.

Kiebel, S.J., von Kriegstein, K., Daunizeau, J., Friston, K.J., 2009. Recognizing sequences of sequences. PLoS Comput Biol 5, e1000464.

Limanowski, J., 2017. (Dis-)Attending to the Body, in: Metzinger, T.K., Wiese, W. (Eds.), Philosophy and Predictive Processing. MIND Group, Frankfurt am Main.

Olshausen, B.A., Field, D.J., 1996. Emergence of simple-cell receptive field properties by learning a sparse code for natural images. Nature 381, 607-609.

Rao, R.P., 1999. An optimal estimation approach to visual perception and learning. Vision Res 39, 1963-1989.

Rao, R.P.N., Ballard, D.H., 1999. Predictive coding in the visual cortex: a functional interpretation of some extra-classical receptive-field effects. Nature Neuroscience 2, 79-87.

Srinivasan, M.V., Laughlin, S.B., Dubs, A., 1982. Predictive coding: a fresh view of inhibition in the retina. Proc R Soc Lond B Biol Sci 216, 427-459.

Reviewer #2: Uploaded

**Have the authors made all data and (if applicable) computational code underlying the findings in their manuscript fully available?**

Reviewer #1: Yes

Reviewer #2: Yes

PLOS authors have the option to publish the peer review history of their article (what does this mean?). If published, this will include your full peer review and any attached files.

Reviewer #1: **Yes: **Karl Friston

Reviewer #2: No
---

## [Decision Letter · Decision Letter 1]

7 Mar 2023

Dear Dr. Bogacz,

We are pleased to inform you that your manuscript 'Recurrent predictive coding models for associative memory employing covariance learning' has been provisionally accepted for publication in PLOS Computational Biology.

Best regards,

Xue-Xin Wei

Academic Editor

PLOS Computational Biology

Daniele Marinazzo

Section Editor

PLOS Computational Biology

Reviewer's Responses to Questions

**Comments to the Authors:**

Reviewer #1: Many thanks for attending to my previous comments – and congratulations on a thoughtful piece of work

Reviewer #2: My issues have been addressed by the authors. I much appreciate the reworking of the introduction and the clarification of concerns related to both the PC and AM theories. Thank you very much.

**Have the authors made all data and (if applicable) computational code underlying the findings in their manuscript fully available?**

Reviewer #1: Yes

Reviewer #2: Yes

PLOS authors have the option to publish the peer review history of their article (what does this mean?). If published, this will include your full peer review and any attached files.

Reviewer #1: No

Reviewer #2: No

---

## [Editor Report · Acceptance letter]

6 Apr 2023

PCOMPBIOL-D-22-01633R1 

Recurrent predictive coding models for associative memory employing covariance learning

Dear Dr Bogacz,

I am pleased to inform you that your manuscript has been formally accepted for publication in PLOS Computational Biology. Your manuscript is now with our production department and you will be notified of the publication date in due course.

With kind regards,

Zsofi Zombor
